# MEMTO: Memory-guided Transformer for Multivariate Time Series Anomaly Detection

**Junho Song**[1*] **Keonwoo Kim**[1,2*] **Jeonglyul Oh**[1] **Sungzoon Cho**[1†]

[1]Seoul National University
[2]VRCREW Inc.
{jhsong, keonwookim, jamesoh0813}@bdai.snu.ac.kr
zoon@snu.ac.kr

## Abstract

Detecting anomalies in real-world multivariate time series data is challenging due to complex temporal dependencies and inter-variable correlations. Recently, reconstruction-based deep models have been widely used to solve the problem. However, these methods still suffer from an over-generalization issue and fail to deliver consistently high performance. To address this issue, we propose the MEMTO, a memory-guided Transformer using a reconstruction-based approach. It is designed to incorporate a novel memory module that can learn the degree to which each memory item should be updated in response to the input data. To stabilize the training procedure, we use a two-phase training paradigm which involves using $K$-means clustering for initializing memory items. Additionally, we introduce a bi-dimensional deviation-based detection criterion that calculates anomaly scores considering both input space and latent space. We evaluate our proposed method on five real-world datasets from diverse domains, and it achieves an average anomaly detection F1-score of 95.74%, significantly outperforming the previous state-of-the-art methods. We also conduct extensive experiments to empirically validate the effectiveness of our proposed model's key components.

## 1 Introduction

As cyber-physical systems advance, a vast amount of time series data is continuously collected from numerous sensors. Anomalies resulting from malfunctions in critical infrastructures, such as water treatment facilities and space probes, can incur fatal property loss. The task of multivariate time series anomaly detection involves identifying whether each timestamp of the multivariate time series is normal or abnormal. Anomaly detection in real-world scenarios is challenging due to severe data imbalance and the prevalence of unlabeled anomalies. We have formulated the problem as an unsupervised learning task to tackle these challenges. The underlying assumption of this approach is that the training data solely consists of normal samples [25, 42].

Traditional unsupervised learning methods such as one-class SVM (OC-SVM) [28], support vector data description (SVDD) [35], isolation forest [20], and local outlier factor (LOF) [5] have been widely used for anomaly detection tasks. Recently, density-estimation methods combined with deep representation learning, such as DAGMM [45] and MPPCACD [41], have also been presented. For the clustering-based method, Deep SVDD [25] finds the smallest hyper-sphere enclosing most normal samples in the feature space trained using deep neural networks. However, they exhibit poor performance in the time series domain due to their inability to capture dynamic nonlinear temporal

---

[*]indicates equal contributions
[†]indicates corresponding author

37th Conference on Neural Information Processing Systems (NeurIPS 2023).

dependencies and complex inter-variable correlations. Deep models tailored for sequential data have been introduced to solve these inherent challenges [30, 7, 29]. THOC [29] uses a differentiable hierarchical clustering mechanism to combine temporal features of varying scales from different resolutions. Multiple hyper-spheres are used to represent each normal pattern at various resolutions, improving the representational capacity to capture intricate temporal features of time series data.

One of the primary types of recent deep methods is a reconstruction-based method, which uses an encoder-decoder architecture trained by a self-supervised pretext task of reconstructing input. This approach expects accurate reconstruction for normal samples and high reconstruction errors for anomalies. Early methods include LSTM-based encoder-decoder model [22] and LSTM-VAE [23]. OmniAnomaly [31] and InterFusion [19] are other stochastic recurrent neural network-based models extended from LSTM-VAE. Another branch of the reconstruction-based method is deep generative models. MAD-GAN [17] and BeatGAN [42] are generative adversarial network [9] variants tailored for time series anomaly detection. The most recently proposed Anomaly Transformer [40] introduces the Anomaly-Attention mechanism to simultaneously model prior- and series- associations. However, to the best of our knowledge, existing reconstruction-based methods may suffer from an over-generalization problem, where abnormal inputs are reconstructed too well [8, 24]. This can occur if the encoder extracts unique features of an anomaly or if the decoder has excessive decoding capabilities for abnormal encoding vectors.

Our paper presents a new reconstruction-based method, a memory-guided Transformer, for multivariate time series anomaly detection (**MEMTO**). One of the key components in MEMTO is the Gated memory module, which includes items representing the prototypical features of normal patterns in the data. We employ an incremental approach to train the individual items in the Gated memory module. It determines the degree to which each existing item should be updated based on the input data. This approach enables MEMTO to adjust to diverse normal patterns in a more data-driven manner. Reconstructing abnormal samples using the stored features of normal patterns in the memory items can lead to reconstruction outputs that resemble normal samples. Under this condition, MEMTO finds it difficult to reconstruct anomalies, thereby releasing the over-generalization issue. Moreover, we have noticed that updating memory items incrementally can result in training instability if the items are initialized randomly. Therefore, we propose a two-phase training paradigm to ensure stable training by injecting inductive bias of normal prototypical patterns into memory items through applying $K$-means clustering. Furthermore, our observation reveals that the latent representations of anomalies are considerably more distant from memory items than those of normal time points since each item encompasses a prototype of normal patterns. Consequently, while calculating anomaly scores, we attempt to fully employ the memory items' nature of storing prototypical features of normal patterns. We introduce a bi-dimensional deviation-based detection criterion that comprehensively considers both the input and latent space. Our approach has been tested on five standard benchmarks, which include real-world applications. Experimental results indicate that our approach is effective and achieves state-of-the-art anomaly detection results compared to existing methods. The contributions of this paper are fourfold:

- Our proposed MEMTO is the first multivariate time series anomaly detection method that uses the Gated memory module, which adjusts to diverse normal patterns in a data-driven manner.
- We propose a two-phase training paradigm, an universally applicable approach designed to enhance the stability and robustness of memory module-based models.
- We propose a bi-dimensional deviation-based detection criterion in the online detection process. It considers both latent and input space to aggregate information in data comprehensively.
- MEMTO achieves state-of-the-art results on five real-world benchmark datasets. Extensive ablation studies demonstrate the effectiveness of key components in MEMTO.

## 2   Related work

In this section, we provide a brief explanation of the memory network. [11, 38] are early studies of external memory that can be retrieved and written. [34] proposes the improved version of Memory Networks [38], which uses attention mechanism [3] to retrieve memory entries, thus requiring less training supervision due to an end-to-end training manner. Recently, several attempts have been

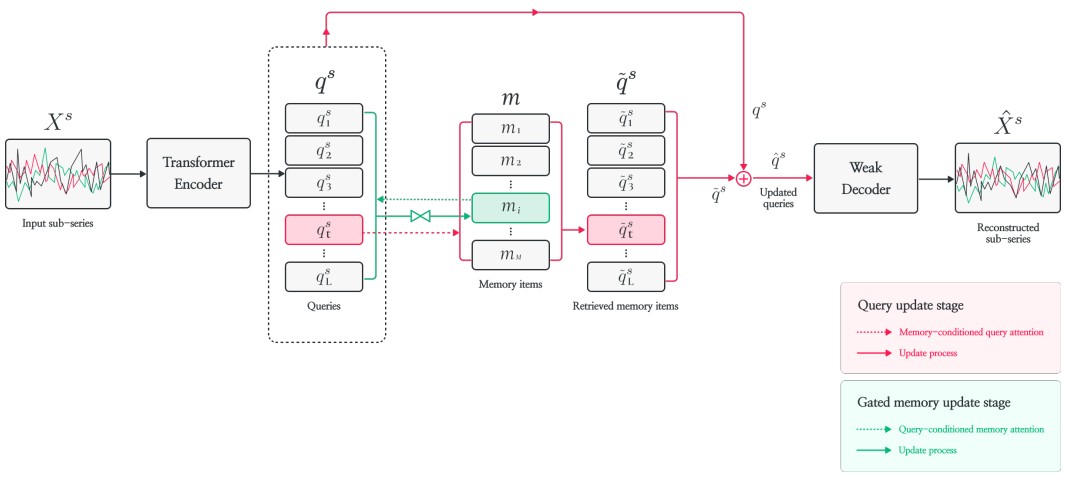

Figure 1: Illustration of proposed MEMTO. Green $\bowtie$ depicts the update gate $\psi$ in the Gated memory update stage, which controls the degree of updating items. Red $\oplus$ represents concatenation of queries $q_t^s$ and retrieved memory items $\tilde{q}_t^s$. 'Weak Decoder' represents two fully-connected layers.

made to apply memory network in various domains. Natural language processing includes question answering [16, 39, 11, 34]. In computer vision, there exist video representation learning [12], one-shot learning [14, 6, 27], and text-to-image synthesis [44]. Moreover, several works have also adopted memory network for anomaly detection in computer vision [8, 24, 21]. Although MemAE [8] is the first to integrate a memory network into autoencoder architecture, it lacks an explicit memory updating process. MNAD [24] has suggested a distinct strategy for updating memory, which involves explicitly storing diverse normal patterns of normal data in memory items. Nonetheless, this approach simply adds a weighted sum of the relevant queries to each memory item without controlling the degree of new information being injected into existing memory items. However, our Gated memory module in MEMTO is the first to adapt to diverse normal patterns in a data-driven way by learning how intensively each existing memory item should be updated in response to new normal patterns.

## 3 Method

### 3.1 Overall description of MEMTO

We define the raw time series $D$ as a set of sub-series $D = \{X^1, \ldots, X^N\}$, where $N$ represents the total number of sub-series. Each sub-series $X^s \in \mathbb{R}^{L \times n}$ is a sequence of timestamps $[x_1^s, \ldots, x_L^s]$. $L$, $n$, and $x_t^s \in \mathbb{R}^n$ represent sub-series length, input dimension, and timestamp at time $t$, respectively. $q^s = [q_1^s, \ldots, q_L^s] \in \mathbb{R}^{L \times C}$ refers to the encoder output feature (i.e., a query sequence) corresponding to $X^s$, where $C$ denotes a latent dimension. $q_t^s \in \mathbb{R}^C$ ($t = 1, \ldots, L$) is an individual query vector (i.e., a query) at time $t$.

Figure 1 illustrates the overall architecture of MEMTO, which mainly consists of encoder-decoder architecture with the Gated memory module. The input sub-series $X^s$ is first fed to the encoder, and the encoder output features are then used as queries to retrieve relevant items or update items in the Gated memory module. The inputs of the decoder are the updated queries $\hat{q}_t^s \in \mathbb{R}^{2C}$ ($t = 1, \ldots, L$), which are the combined features of the queries $q_t^s$ and the retrieved memory items $\tilde{q}_t^s \in \mathbb{R}^C$. The decoder maps the updated query sequence $\hat{q}^s = [\hat{q}_1^s, \ldots, \hat{q}_L^s]$ back to the input space, and outputs the reconstructed input sub-series $\hat{X}^s \in \mathbb{R}^{L \times n}$. Algorithm 2 shows the overall mechanism for our proposed model in Appendix B.

#### 3.1.1 Encoder and decoder

Transformer [36] is well known to capture long-term complex temporal patterns in time series data [43]. MEMTO uses a Transformer encoder to project the input sub-series into a latent space.

In contrast, we adopt a weak decoder consisting of two fully-connected layers. It is essential that the decoder should not be overly powerful, as this can lead to a situation where its performance is independent of the encoder's ability to encode input time series. In the extreme case, a strong decoder comprised of multiple deep layers can generate the input data accurately even from random noise that contains no information about input data [37].

### 3.1.2 Gated memory module

In order to tackle the problem of over-generalization in reconstruction-based models, we introduce a new memory module mechanism that adjusts to diverse normal patterns in a data-driven manner. In this approach, each item stored in the memory module represents the prototypical features of normal data. Our proposed two-stage iterative update process allows us to extract latent representations of input sub-series using the prototypical patterns of normal data stored in the memory module, which acts as regularization to mitigate over-generalization. This mechanism is designed to limit the encoder's ability to capture the unique properties of anomalies, thereby making it more challenging to reconstruct abnormal data.

**Gated memory update stage** Memory item $m_i \in \mathbb{R}^C$ ($i = 1, \ldots, M$), where $M$ denotes the number of memory items, is trained to contain prototypical normal patterns of the input time series [24]. We expect memory items to contain prototypes of all queries $q_t^s$ corresponding to normal timestamps. Thus, we adopt an incremental approach for updating the memory items by defining query-conditioned memory attention $v_{i,t}^s$. It is calculated by softmax on dot product between each memory item and queries as follows:

$$v_{i,t}^s = P(m_i \rightarrow q_t^s) = \frac{exp(\langle m_i, q_t^s \rangle / \tau)}{\sum_{p=1}^{L} exp(\langle m_i, q_p^s \rangle / \tau)}, \tag{1}$$

where $\tau$ denotes a temperature hyperparameter. We have proposed an update gate $\psi$ in our memory module mechanism to train each memory item flexibly according to various normal patterns. This gate controls the extent to which the newly acquired normal patterns from queries are infused into the existing prototypical normal patterns stored in memory items. It allows our model to learn to what degree each existing memory item should be updated in a data-driven way. Equations for our memory update mechanism are:

$$\psi = \sigma \left( U_\psi m_i + W_\psi \sum_{t=1}^{L} v_{i,t}^s q_t^s \right), \tag{2}$$

$$m_i \leftarrow (1 - \psi) \circ m_i + \psi \circ \sum_{t=1}^{L} v_{i,t}^s q_t^s, \tag{3}$$

where $U_\psi$ and $W_\psi$ denote linear projections, and $\sigma$ and $\circ$ denotes sigmoid activation and element-wise multiplication, respectively. The memory update stage is only executed in the training phase.

**Query update stage** In query update stage, we generate updated queries $\hat{q}_t^s$ which are then fed to decoder as inputs. Similar to the Gated memory update stage, we define memory-conditioned query-attention $w_{t,i}^s$, calculated by softmax on dot product between each query and memory items as follows:

$$w_{t,i}^s = P(q_t^s \rightarrow m_i) = \frac{exp(\langle m_i, q_t^s \rangle / \tau)}{\sum_{j=1}^{M} exp(\langle m_j, q_t^s \rangle / \tau)}, \tag{4}$$

Then, retrieved memory item $\tilde{q}_t^s$ is obtained by weighted sum of the items $m_i$, using $w_{t,i}^s$ as its corresponding weight as follows:

$$\tilde{q}_t^s = \sum_{i'=1}^{M} w_{t,i'}^s m_{i'}. \tag{5}$$

Query $q_t^s$ and retrieved memory item $\tilde{q}_t^s$ are concatenated along feature dimension to constitute updated query $\hat{q}_t^s$. The updated queries $\hat{q}_t^s$ $(t = 1, \ldots, L)$ are the new robust inputs for the decoder since unique properties of an anomaly in $q_t^s$ can be offset by the relevant normal patterns in the memory items. Reconstruction outputs of anomalies often appear similar to normal samples, making it more challenging to reconstruct anomalies. Such intensified difficulty can help distinguish between normal and abnormal data more effectively and prevent over-generalization.

## 3.2 Training

For the pretext task for self-supervision, we minimize reconstruction loss while training. The reconstruction loss $L_{rec}$ is defined as $L2$ loss between $X^s$ and $\hat{X}^s$:

$$L_{rec} = \frac{1}{N} \sum_{s=1}^{N} \left\| X^s - \hat{X}^s \right\|_2^2. \tag{6}$$

Dense $W^s$ enables some anomalies to have the probability of being well reconstructed [8], where $W^s$ denotes the matrix form of $w_{t,i}^s$ $(t = 1, \ldots, L)$ and $(i = 1, \ldots, M)$ in (4). Hence, to guarantee retrieving only a restricted number of closely relevant normal prototypes in memory, we introduce entropy loss $L_{entr}$ as our auxiliary loss for sparsity regularization on $W^s$:

$$L_{entr} = \frac{1}{N} \sum_{s=1}^{N} \sum_{t=1}^{L} \sum_{i=1}^{M} -w_{t,i}^s log(w_{t,i}^s). \tag{7}$$

The objective function $L$ is to minimize the combination of loss term (6) and (7) as follows:

$$L = L_{rec} + \lambda L_{entr}, \tag{8}$$

where $\lambda$ denotes a weighting coefficient.

## 3.3 Two-phase training paradigm

---
**Algorithm 1** Memory module initialization with $K$-means clustering

---
**Input:** $X \in \mathbb{R}^{N_{rand} \times L \times n}$: randomly sampled sub-series
      $\tilde{f}_e$: trained encoder of MEMTO after first phase
      $m \in \mathbb{R}^{M \times C}$: randomly initialized memory items
      ($N_{rand}$: number of sub-series, $M$: number of memory items, $C$: latent dimension)
1:  $q^{rand} = \tilde{f}_e(X)$       $\backslash\backslash$ $q^{rand} \in \mathbb{R}^{T \times C}$, where $T = N_{rand} \times L$
2:  $c = K\text{-means}(q^{rand})$     $\backslash\backslash$ Get $M$ clustering centroids $c \in \mathbb{R}^{M \times C}$
3:  **for** $i = 1, \ldots, M$ **do**     $\backslash\backslash$ Initialize memory items $m$ with clustering centroids $c$
4:     $m_i = c_i$
5:  **end for**
6:  **return** $m$

---

**Initializing memory items with $K$-means clustering**   Since we update the memory items incrementally, there is a risk of instability during training if the items are randomly initialized. We suggest a novel two-phase training paradigm that uses a clustering method to set the initial value of memory items to the approximate normal prototypical pattern of the data.

In the first phase, MEMTO is trained by a self-supervised task of reconstructing inputs, and the trained encoder of MEMTO generates queries for the randomly sampled 10% of the training data. We then apply $K$-means clustering algorithm to cluster the queries and designate each centroid as the initial value of a memory item. In the second phase, MEMTO is trained with these well-initialized items on anomaly detection tasks. Algorithm 1 outlines the memory module initialization with $K$-means clustering. In all of our experiments, we use $K$-means clustering as a baseline clustering method. Our claim is not that $K$-means clustering is the optimal choice among various clustering

methods, but rather the effectiveness of a two-phase training paradigm that allows for the selection of an appropriate clustering algorithm depending on the type of dataset.

## 3.4 Anomaly criterion

We introduce a bi-dimensional deviation-based detection criterion that comprehensively considers input and latent space. We define Latent Space Deviation (LSD) $LSD(q_t^s, m)$ at time point $t$ as distance between each query $q_t^s$ and its nearest memory item $m_t^{s,pos}$ in latent space (9). LSD of anomalies would be larger than those of normal time points because each memory item contains a prototype of normal patterns. Additionally, we define Input Space Deviation (ISD) $ISD(X_{t,:}^s, \hat{X}_{t,:}^s)$ at time $t$ as distance between input $X_{t,:}^s \in \mathbb{R}^n$ and reconstructed input $\hat{X}_{t,:}^s \in \mathbb{R}^n$ in input space (10).

$$LSD(q_t^s, m) = \|q_t^s - m_t^{s,pos}\|_2^2 \tag{9}$$

$$ISD(X_{t,:}^s, \hat{X}_{t,:}^s) = \left\| X_{t,:}^s - \hat{X}_{t,:}^s \right\|_2^2 \tag{10}$$

We multiply normalized LSD with ISD, using LSD as weights for amplifying the normal-abnormal gap in ISD:

$$A(X^s) = softmax\left([LSD(q_t^s, m)]_{t=1,...,L}\right) \circ [ISD(X_{t,:}^s, \hat{X}_{t,:}^s)]_{t=1,...,L}, \tag{11}$$

where $\circ$ is element-wise multiplication and $A(X^s) \in \mathbb{R}^L$ is anomaly score at each time point. Leveraging normal-abnormal distinguishing criteria in latent space (i.e., Latent Space Deviation) and input space (i.e., Input Space Deviation) leads to better detection performance.

# 4 Experiment

Table 1: Precision(P), recall(R), F1-score(F1) results (as %) on five real-world datasets. 'A.T.' and 'avg' indicate Anomaly Transformer and average. We reproduce the MEMTO and Anomaly Transformer results while adopting the reported performance from [40] for the other baselines.

| | SMD | | | MSL | | | SMAP | | | SWaT | | | PSM | | | avg. |
|---|---|---|---|---|---|---|---|---|---|---|---|---|---|---|---|---|
| | P | R | F1 | P | R | F1 | P | R | F1 | P | R | F1 | P | R | F1 | F1 |
| LOF | 56.34 | 39.86 | 46.68 | 47.72 | 85.25 | 61.18 | 58.93 | 56.33 | 57.60 | 72.15 | 65.43 | 68.62 | 57.89 | 90.49 | 70.61 | 60.94 |
| OC-SVM | 44.34 | 76.72 | 56.19 | 59.78 | 86.87 | 70.82 | 53.85 | 59.07 | 56.34 | 45.39 | 49.22 | 47.23 | 62.75 | 80.89 | 70.67 | 60.25 |
| IsolationForest | 42.31 | 73.29 | 53.64 | 53.94 | 86.54 | 66.45 | 52.39 | 59.07 | 55.53 | 49.29 | 44.95 | 47.02 | 76.09 | 92.45 | 83.48 | 61.22 |
| MMPCACD | 71.20 | 79.28 | 75.02 | 81.42 | 61.31 | 69.95 | 88.61 | 75.84 | 81.73 | 82.52 | 68.29 | 74.73 | 76.25 | 78.35 | 77.29 | 75.74 |
| DAGMM | 67.30 | 49.89 | 57.30 | 89.60 | 63.93 | 74.62 | 86.45 | 56.73 | 68.51 | 89.92 | 57.84 | 70.40 | 93.49 | 70.03 | 80.08 | 70.18 |
| Deep-SVDD | 78.54 | 79.67 | 79.10 | 91.92 | 76.63 | 83.58 | 89.93 | 56.02 | 69.04 | 80.42 | 84.45 | 82.39 | 95.41 | 86.49 | 90.73 | 80.97 |
| THOC | 79.76 | 90.95 | 84.99 | 88.45 | 90.97 | 89.69 | 92.06 | 89.34 | 90.68 | 83.94 | 86.36 | 85.13 | 88.14 | 90.99 | 89.54 | 88.01 |
| LSTM-VAE | 75.76 | 90.08 | 82.30 | 85.49 | 79.94 | 82.62 | 92.20 | 67.75 | 78.10 | 76.00 | 89.50 | 82.20 | 73.62 | 89.92 | 80.96 | 81.24 |
| BeatGAN | 72.90 | 84.09 | 78.10 | 89.75 | 85.42 | 87.53 | 92.38 | 55.85 | 69.61 | 64.01 | 87.46 | 73.92 | 90.30 | 93.84 | 92.04 | 80.24 |
| OmniAnomaly | 83.68 | 86.82 | 85.22 | 89.02 | 86.37 | 87.67 | 92.49 | 81.99 | 86.92 | 81.42 | 84.30 | 82.83 | 88.39 | 74.46 | 80.83 | 84.69 |
| InterFusion | 87.02 | 85.43 | 86.22 | 81.28 | 92.70 | 86.62 | 89.77 | 88.52 | 89.14 | 80.59 | 85.58 | 83.01 | 83.61 | 83.45 | 83.52 | 85.70 |
| A.T. | 87.96 | 94.68 | 91.20 | 91.13 | 90.12 | 90.63 | 93.96 | 98.45 | 96.15 | 85.85 | 100.00 | 92.39 | 96.81 | 98.63 | 97.71 | 93.62 |
| MEMTO | 89.13 | 98.40 | 93.54 | 92.07 | 96.76 | 94.36 | 93.76 | 99.63 | 96.61 | 94.18 | 97.54 | 95.83 | 97.46 | 99.23 | 98.34 | 95.74 |

## 4.1 Experimental setup

**Dataset** We evaluate MEMTO on five real-world multivariate time series datasets. (i) Server Machine Dataset (SMD [33]) is a large-scale dataset collected over five weeks with 38 dimensions released from a large Internet company. (ii & iii) Mars Science Laboratory rover (MSL) and Soil Moisture Active Passive satellite (SMAP) are public data released from NASA [13], with 55 and 38 dimensions, respectively. (iv) Secure Water Treatment (SWaT [18]) consists of a six-stage infrastructure process with 51 sensors under 11 days of continuous operation. (v) Pooled Server Metrics (PSM [1]) consists of data of diverse application server nodes from eBay with 26 dimensions. Additional details on the datasets can be found in Appendix A.

**Implementation details**   We generate sub-series by applying a non-overlapped sliding window with a length of 100 to obtain fixed-length inputs for each dataset. We split the training data into 80% for training and 20% for validation. More detailed information on hyperparameter settings can be found in Appendix A. The standard evaluation metrics for binary classification, including precision, recall, and F1-score, fail to account for the sequential properties of time series data, making them insufficient in assessing contextual and collective anomalies [10, 4]. Therefore, we use adjusted versions of these metrics, which have become widely used evaluation metrics in time series anomaly detection [32, 2]. Here point adjustment method is used, where if a single timestamp is detected as abnormal, every time point in the anomalous segment containing that timestamp is considered correctly detected.

## 4.2   Main results

In the main experiment, we evaluate the performance of MEMTO on multivariate time series anomaly detection tasks by comparing it with 12 models. The traditional machine learning baselines are LOF [5], OC-SVM [28], and Isolation Forest [20]. We also compare with several recent deep models, including density-estimation models (MPPCAD [41] and DAGMM [45]), clustering-based models (Deep-SVDD [26] and THOC [29]), and reconstruction-based models (LSTM-VAE [23], BeatGAN [42], OmniAnomaly [31], InterFusion [19], and Anomaly Transformer [40]).

Table1 shows the evaluation results on five multivariate time series anomaly detection tasks. Overall, models using Transformers such as MEMTO and Anomaly Transformer consistently show high performance with an F1-score of over 90% on all datasets. However, MEMTO substantially improves the average F1-score on benchmarks compared to the previous state-of-the-art model, Anomaly Transformer, from 93.62% to 95.74%. MEMTO effectively detects anomalies in complex time series data by adjusting to diverse normal patterns present in the data and creating prototypes of these patterns using a data-driven approach, thus achieving new state-of-the-art results.

## 4.3   Ablation study

In this section, we comprehensively analyze the proposed model, MEMTO, through a series of ablation studies on three key components: the anomaly criterion, the memory module, and the training paradigm. Appendix C shows a more detailed analysis of the ablation studies.

Table 2: Effectiveness of ISD and LSD.

| Anomaly Criterion | | F1-score | | | | | |
|---|---|---|---|---|---|---|---|
| ISD | LSD | SMD | MSL | PSM | SMAP | SWaT | avg. |
| ✓ | ✗ | 77.54 | 87.22 | 79.25 | 70.99 | 31.17 | 69.23 |
| ✗ | ✓ | 72.78 | 80.33 | 80.15 | 67.55 | 0.00 | 60.16 |
| ✓ | ✓ | **93.54** | **94.36** | **98.34** | **96.61** | **95.83** | **95.73** |

Table 3:  Ablation results of memory module and training paradigm. 'MM' denotes memory module.

| | | F1-score | | | | | |
|---|---|---|---|---|---|---|---|
| | $K$-means | SMD | MSL | PSM | SMAP | SWaT | avg. |
| w/o MM | - | 77.40 | 87.47 | 79.83 | 71.20 | 31.42 | 69.46 |
| MemAE | ✗ | 89.12 | 90.42 | 86.70 | 70.16 | 61.12 | 79.50 |
| | ✓ | 91.39 | 92.50 | 98.00 | 96.13 | 93.06 | 94.21 |
| MNAD | ✗ | 65.56 | 85.41 | 93.43 | 69.45 | 82.31 | 79.23 |
| | ✓ | 92.39 | 92.13 | 98.13 | 96.04 | 94.52 | 94.64 |
| MEMTO | ✗ | 86.77 | 91.54 | 97.06 | 70.07 | 91.24 | 87.33 |
| | ✓ | **93.54** | **94.36** | **98.34** | **96.61** | **95.83** | **95.73** |

**Anomaly criterion**   Unlike other time series anomaly detection models, MEMTO considers both the input and latent space to derive anomaly scores. Table2 presents the performance of MEMTO

using different anomaly detection criteria. Using only one of the two anomaly criteria, ISD or LSD, the average F1-score is low, by 69.23% and 60.16%, respectively. In particular, they exhibit extremely low performance on SWaT, indicating a high variance in performance across datasets. In contrast, our proposed method, which uses both ISD and LSD, consistently shows the highest performance over all datasets with relatively low variance compared to others.

**Memory module**   In this study, we assess the capability of our novel Gated memory module in anomaly detection tasks. Table 3 shows that removing the Gated memory module results in a significant decrease of the average F1-score by 32.56p% and a remarkable decrease of the F1-score specifically on SWaT to less than a third. To further validate the effectiveness of our approach, we also compare it with two existing memory module mechanisms (MemAE [8] and MNAD [24]) by replacing the memory module in MEMTO while using the same experimental settings. Our findings suggest that simply adding an explicit memory update process does not necessarily improve the performance, as evidenced by the comparable performance between MemAE and MNAD. However, significant performance improvements are observed when we apply our proposed memory module that integrates the update gate into the memory update process. Our approach demonstrates its superiority in anomaly detection tasks on diverse domains of datasets.

**Training paradigm**   In order to enhance the stability of memory item updates during training, we use $K$-means clustering to initialize the values of memory items instead of random initialization. Our experimental results provide strong evidence that supports the effectiveness of the two-phase training paradigm of MEMTO. Table 3 demonstrates that not applying $K$-means clustering results in an 8.4p% decrease in the average F1-score of MEMTO compared to when it is applied. For such cases in MemAE and MNAD, scores decrease by 14.7p% and 15.4p%, respectively, which indicates that our proposed training paradigm can be universally applicable to memory module-based models.

## 4.4   Discussion and analysis

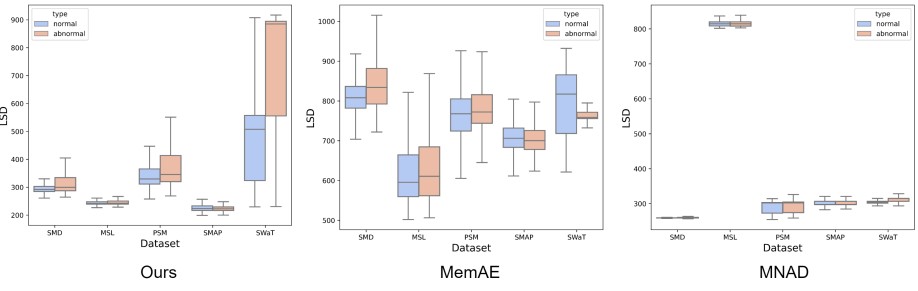

Figure 2: Distribution of LSD values for normal and abnormal samples.

Table 4: Statistical analysis results of LSD. Values in the table are calculated as follows: $\mathbb{E}_{q^+ \in Normal} LSD(q^+, m) / \mathbb{E}_{q^- \in Abnormal} LSD(q^-, m)$, where $q^+$ and $q^-$ denote query of normal and abnormal sample respectively.

|  | **SMD** | **MSL** | **PSM** | **SMAP** | **SWaT** |
|---|---|---|---|---|---|
| MemAE | 0.9674 | 0.9719 | 0.9796 | 1.0059 | 1.0321 |
| MNAD | 1.0052 | 1.0043 | 0.9960 | **1.0004** | 0.9750 |
| Ours | **0.9014** | **0.9484** | **0.9368** | 1.0098 | **0.6240** |

**Statistical analysis of LSD**   We further explore the Gated memory module's superior ability to capture prototypical normal patterns in data. Figure 2 demonstrates box plots depicting the distribution of LSD values for normal and abnormal samples across three memory modules: ours,

MemAE, and MNAD. We calculate mean LSD values for normal and abnormal samples as follows: $\mathbb{E}_{q^+ \in Normal} LSD(q^+, m)$ and $\mathbb{E}_{q^- \in Abnormal} LSD(q^-, m)$, where $m$ denotes fixed memory items. Mean LSD values for each type of memory module are in Appendix D.

We calculate the ratio between the mean LSD values for normal and abnormal samples in the test dataset. Table 4 presents the ratio values for each dataset while using three different memory modules. Our proposed approach consistently shows smaller ratio values than those of other methods across the majority of datasets. This suggests that the items stored in the Gated memory module are more distant from queries of abnormal samples than those of normal samples, and the gap between them is greater than that in other types of memory modules. It supports our assertion that the Gated memory module better encodes normal patterns in the data than other compared methods.

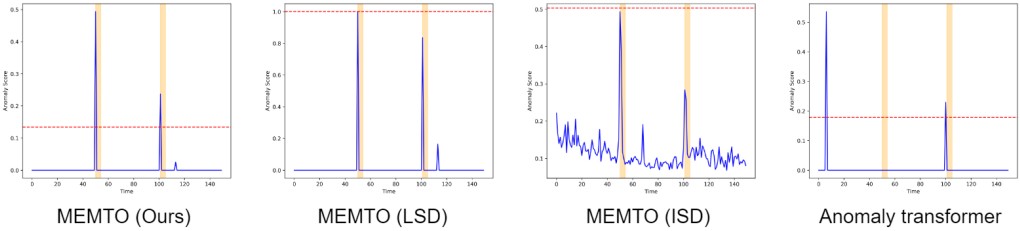

Figure 3: Visualization of anomaly scores on SMD. Each graph displays the anomaly scores on the y-axis and time on the x-axis. The yellow-shaded area indicates the ground truth of abnormal labels, while the blue line represents the anomaly scores predicted by the models, and the red dashed line represents the threshold values. Additional plots for different datasets are included in Appendix D.

**Anomaly score**  We experiment on randomly extracted time segments of length 150 from all datasets to demonstrate the effectiveness of the bi-dimensional deviation-based detection criterion. The baselines used in this experiment include MEMTO leveraging either LSD or ISD as a detection criterion and the previous state-of-the-art model, Anomaly Transformer. Figure 3 illustrates the anomaly score and ground truth label associated with each timestamp within a segment. The results show that the bi-dimensional deviation-based criterion robustly detects anomalies compared to the baselines. Compared to Anomaly Transformer, which uses association-based detection criteria, the bi-dimensional deviation-based criterion shows precise detection with reduced false alarms. Furthermore, if either latent or input space aspects are removed, MEMTO shows unstable performance. While MEMTO using only LSD as a detection criterion exhibits a similar pattern in anomaly scores, scores in abnormal time points fail to exceed the threshold. This suggests that considering both factors is crucial for robust performance since it amplifies the gap between normal and abnormal time points based on anomaly scores.

Table 5: Time (in seconds) measured for each stage. Experiments are done on SWaT dataset.

| | time (sec) | | | | |
|---|---|---|---|---|---|
| | First phase | K-means | Second phase | Total training | Inference |
| MEMTO | 45.17 | 16.07 | 60.44 | 121.7 | 9.360 |
| w/o Gated memory module | 39.04 | - | - | 39.04 | 9.230 |
| w/o two-phase training paradigm | 49.62 | - | - | 49.62 | 9.360 |
| Anomaly Transformer | 46.87 | - | - | 46.87 | 10.53 |

**Computational efficiency**  We measure the training time with and without the use of the Gated memory module and the two-phase training paradigm, respectively. The results are presented in Table 5. When training MEMTO without the Gated memory module, there is a reduction in both the number of parameters and training time. However, this results in a 26.27%p decline in performance, as in indicated in Table 3. Also, the two-phase training paradigm increases the training duration by approximately 2.45 times compared to its absence, though the the inference time remains unaffected.

Remarkably, the two-phase training paradigm enhances performance by 10.4%p, justifying its adoption even with the extended training duration.

We further compare the computational efficiency between MEMTO and the previous state-of-the-art model, Anomaly Transformer. The Anomaly Transformer computes and retains series-association and prior-association per encoder layer, then averages the association discrepancies across layers. Performing this process during inference is computationally demanding, while MEMTO, requiring only dot product operations between each query and a few memory items, is simpler and results in shorter inference time. As presented in Table 5, while MEMTO requires over twice the training time of Anomaly Transformer due to the two-phase training paradigm, it is noteworthy that when considering the critical inference time for real-world applications, MEMTO is faster by 1.17 seconds.

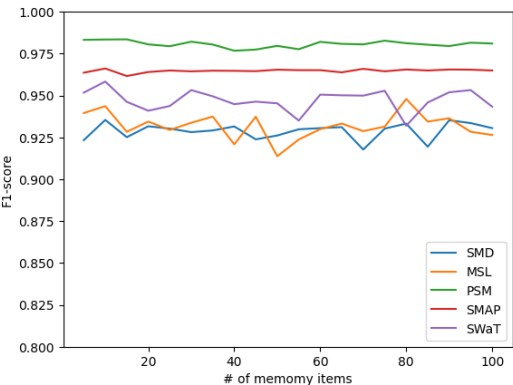

Figure 4: Performance across different numbers of memory items in the Gated memory module

Furthermore, given that the computational cost increases as the number of memory items within MEMTO's Gated memory module grows, we conduct experiments to analyze the optimal number of memory items. Figure 4 illustrates the relationship between the performance of MEMTO's performance and the number of memory items used. Results show that MEMTO's performance is robust to the number of memory items, as the performance variance across datasets is small. Hence, we designate ten memory items as the default value after weighing performance and computational complexity. Our study highlights the effectiveness of employing a restricted number of memory items to extract prototypical features of normal patterns in time series data. Unlike computer vision, which may require thousands of memory items [8], we demonstrate that only ten memory items are necessary for a task in the time series domain.

## 5   Conclusion

We introduce MEMTO, an unsupervised reconstruction-based model for multivariate time series anomaly detection. The Gated memory module in MEMTO adaptively captures the normal patterns in response to the input data, and it can be trained robustly using a two-phase training paradigm. Our proposed anomaly criterion, which comprehensively considers bi-dimensional space, enhances the performance of MEMTO. Extensive experiments on real-world multivariate time series benchmarks validate that our proposed model achieves state-of-the-art performance compared to existing competitive models.

**Limitations**   While our two update stages in the Gated memory module and bi-dimensional deviation-based criterion have led to enhanced performance, a thorough theoretical proof for its efficacy remains to be established. Also, we acknowledge the limitation of our omission of visually inspecting the prototypical normal patterns stored in memory items. In the future, we will explore these issues further as a part of our research.

**Broader impacts**   MEMTO is tailored for detecting anomalies in multivariate time series data and can be applied to various complex cyber-physical systems, such as smart factories, power grids, data centers, and vehicles. However, we strongly discourage its use in activities related to financial crimes or other applications that could have negative societal consequences.

## Acknowledgements

This work was supported by the National Research Foundation of Korea(NRF) grant funded by the Korea government(MSIT). (No. 2021R1A2C2093785)

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

# A  Training details

We use ten memory items for our MEMTO model, corresponding to the number of clusters in our $K$-means clustering. To determine anomalies, we set the threshold as the top-$p$% of the combined results of the anomaly scores from both the training and validation data, with specified values of $p$ for each dataset outlined in Table 6, following [40]. We set $\lambda$ in the objective function to 0.01, use Adam optimizer [15] with a learning rate of 5e-5, and employ early stopping with the patience of 10 epochs against the validation loss during training. Our experiments are conducted using the Pytorch framework on four NVIDIA GTX 1080 Ti 12GB GPUs. Furthermore, during the execution of our experiment, we make partial references to the code of [40].

## A.1  Hyperparameter settings

Important hyperparameters of MEMTO were determined through grid search, while others were set to commonly used default values based on empirical observations. We performed a grid search to determine the values of each hyperparameter within the following range:

- $\lambda \in \{$1e+0, 5e-1, 1e-1, 5e-2, 1e-2, 5e-3, 1e-3$\}$
- $lr \in \{$1e-4, 3e-4, 5e-4, 1e-5, 3e-5, 5e-5$\}$
- $\tau \in \{$0.1, 0.3, 0.5, 0.7, 0.9$\}$
- $M \in \{$5, 10, 15, 20, 25, 30, 35, 40, 45, 50, 55, 60, 65, 70, 75, 80, 85, 90, 95, 100$\}$

, where $lr$, $\tau$, and $M$ denote the learning rate, the temperature in the softmax function, and the number of clusters, respectively. Since we set the centroids of clusters as memory items, the number of memory items and that of clusters are the same. We set the optimal hyperparameters as follows: $\lambda$ as 1e-2, $lr$ as 5e-5, $\tau$ as 0.1, and $M$ as 10. All experiments in this paper are conducted using the same hyperparameters regardless of the dataset.

## A.2  Dataset

Table 6: Details in five benchmarks. The number of samples in the training, validation, and test sets is represented in the columns labeled 'Train,' 'Valid,' and 'Test,' respectively. The '$p$%' column indicates the anomaly ratio used in the experiment. The 'Dim' column shows the dimension size of the data for each dataset.

|        | Train   | Valid   | Test    | $p(\%)$ | Dim |
|--------|---------|---------|---------|---------|-----|
| SMD    | 566,724 | 141,681 | 708,420 | 0.5     | 38  |
| MSL    | 46,653  | 11,664  | 73,729  | 1.0     | 55  |
| PSM    | 105,984 | 26,497  | 87,841  | 1.0     | 26  |
| SMAP   | 108,146 | 27,037  | 427,617 | 1.0     | 25  |
| SWaT   | 396,000 | 99,000  | 449,919 | 0.1     | 53  |

Table 6 shows the statistical details of datasets used in experiments. We obtained SWaT by submitting a request through `https://itrust.sutd.edu.sg/itrust-labs_datasets/`.

# B   Algorithm for MEMTO

---

**Algorithm 2** Proposed Method **MEMTO**

---

**Input** $X^s \in \mathbb{R}^{L \times n}$: input sub-series
**Training params** $f_e$: encoder, $f_d$: decoder, $U_\psi, W_\psi \in \mathbb{R}^{C \times C}$: linear projection matrices
  1: $q^s = f_e\left(X^s\right)$ \qquad \\ feed-forward encoder, $q^s \in \mathbb{R}^{L \times C}$
  2: $v^s = softmax\left(m\left(q^s\right)^T\right)$ \qquad \\ Gated memory update start, $m \in \mathbb{R}^{M \times C}$, $v^s \in \mathbb{R}^{M \times L}$
  3: $\psi = sigmoid\left(mU_\psi + \left(v^s q^s\right)W_\psi\right)$ \qquad \\ $\psi \in \mathbb{R}^{M \times C}$
  4: $m = \left(1 - \psi\right) \circ m + \psi \circ \left(v^s q^s\right)$ \qquad \\ Gated memory update end
  5: $w^s = softmax\left(q^s\left(m\right)^T\right)$ \qquad \\ Query update start, $w^s \in \mathbb{R}^{L \times M}$
  6: $\tilde{q}^s = w^s m$ \qquad \\ $\tilde{q}^s \in \mathbb{R}^{L \times C}$
  7: $\hat{q}^s = concat\left(\left[q^s, \tilde{q}^s\right], dim = 1\right)$ \qquad \\ Query update end, $\hat{q}^s \in \mathbb{R}^{L \times 2C}$
  8: $\hat{X}^s = f_d\left(\hat{q}^s\right)$ \qquad \\ feed -forward decoder, $\hat{X}^s \in \mathbb{R}^{L \times n}$
  9: **return** $\hat{X}^s$ \qquad \\ reconstructed sub-series

---

Algorithm 2 provides an overall mechanism for our model. It demonstrates the matrix operation version of the forward process when a single input sub-series $X^s$ is fed to MEMTO.

# C   Additional experiments

Table 7: The ablation results (F1-score) in anomaly criterion and objective function. $L_{rec}$ and $L_{entr}$ signify Reconstruction Loss and Entropy Loss, respectively.

| Loss | Anomaly Criterion | | F1-score | | | | | |
|---|---|---|---|---|---|---|---|---|
| | ISD | LSD | SMD | MSL | PSM | SMAP | SWaT | avg. |
| $L_{rec}$ | ✓ | ✗ | 79.63 | 86.23 | 82.15 | 71.18 | 31.29 | 70.09 |
| | ✗ | ✓ | 69.73 | 72.63 | 93.07 | 67.69 | 82.50 | 77.12 |
| | ✓ | ✓ | 93.19 | 92.66 | 98.05 | 96.48 | 93.34 | 94.74 |
| $L_{entr}$ | ✓ | ✗ | 75.71 | 88.39 | 87.47 | 69.28 | 79.28 | 80.02 |
| | ✗ | ✓ | 12.53 | 84.34 | 76.49 | 68.17 | 83.52 | 65.01 |
| | ✓ | ✓ | 88.43 | 93.40 | 97.97 | 96.22 | 92.77 | 93.75 |
| $L_{rec} + \lambda L_{entr}$ | ✓ | ✗ | 77.54 | 87.22 | 79.25 | 70.99 | 31.17 | 69.23 |
| | ✗ | ✓ | 72.78 | 80.33 | 80.15 | 67.55 | 0.00 | 60.16 |
| | ✓ | ✓ | **93.54** | **94.36** | **98.34** | **96.61** | **95.83** | **95.73** |

Table 8: We report mean and standard deviation over 10 runs for A.T (Anomaly Transformer) and MEMTO, respectively. We conduct t-test (p < 0.05) to indicate statistical signficance.

| | | SMD | MSL | SMAP | SWaT | PSM |
|---|---|---|---|---|---|---|
| A.T | mean | 91.34 | 92.60 | 95.83 | 92.86 | 97.57 |
| | std | 0.6942 | 1.103 | 1.088 | 0.9455 | 0.1230 |
| MEMTO | mean | 93.05 | 94.07 | 96.49 | 95.23 | 98.15 |
| | std | 0.3762 | 0.5185 | 0.07987 | 1.016 | 0.1666 |
| p-value | | 0.0000 | 0.0301 | 0.0360 | 0.0002 | 0.0000 |

## C.1   Objective function and anomaly criterion

In this experiment, we investigate the impact of loss terms, specifically the reconstruction loss $L_{rec}$ and the entropy loss $L_{entr}$, on the performance of our proposed framework, MEMTO. We remove

one of the two terms one by one from the objective function and evaluate the resulting performance. Table 7 demonstrates the significance of incorporating both $L_{rec}$ and $L_{entr}$ terms in the objective function. Applying bi-dimensional deviation-based criterion to the MEMTO variants that only use $L_{rec}$ or $L_{entr}$ as the loss function shows competitive performance compared to ours in terms of average F1-score. This demonstrates the robustness of MEMTO to loss terms. Additionally, both cases show a significant performance drop when using only ISD or LSD as the anomaly criterion, emphasizing the importance of combining ISD and LSD for achieving optimal performance.

## C.2    Statistical significance test

We also perform statistical tests comparing our results to the latest state-of-the-art model, Anomaly Transformer. We conduct a t-test to prove a significant performance difference between MEMTO and Anomaly Transformer. The results in Table 8 show a p-value less than 0.05 across all datasets, confirming a significant performance difference between the two models.

## C.3    Number of decoder layers

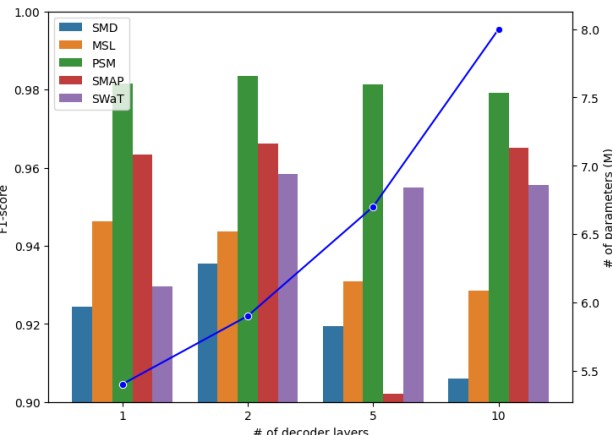

Figure 5: F1-score and number of parameters, according to the number of decoder layers. The right y-axis represents the values of the blue line graph in million units, while the left y-axis represents the values of the bar graph.

Figure 5 provides the performance of MEMTO under different numbers of decoder layers. As shown in Figure 5, a decoder that is too shallow (e.g., a decoder with a single layer) performs worse because it lacks sufficient capacity to reconstruct the input data accurately. On the other hand, if the decoder is too large (e.g., decoder with ten layers), it can become overly expressive and reconstruct even anomalies regardless of the encoding ability of the encoder. Therefore, it can lead to an over-generalization problem, which can ultimately decrease the performance of anomaly detection by reconstructing anomalies too accurately. Furthermore, a larger decoder layer with more parameters can increase computational and memory costs. We empirically find that considering the balance between performance and resource cost, a decoder with two layers is most suitable for anomaly detection tasks presented in our paper.

# D    Additional details for discussion

## D.1    LSD values

Table 9 shows mean LSD values of normal and abnormal samples across various domains of datasets while using different memory module mechanisms. In most datasets, our proposed Gated memory module consistently exhibits a lower mean LSD value for normal samples than for abnormal samples. Furthermore, the relative difference between these values is more significant than other memory module mechanisms. These results demonstrate the efficacy of our memory module mechanism in capturing prototypical features of normal patterns in data.

Table 9: The mean LSD values corresponding to test data.

| | SMD | | MSL | | PSM | | SMAP | | SWaT | |
|---|---|---|---|---|---|---|---|---|---|---|
| | Normal | Abnormal | Normal | Abnormal | Normal | Abnormal | Normal | Abnormal | Normal | Abnormal |
| MemAE | 814.7836 | 842.2023 | 622.5195 | 640.4954 | 766.2473 | 782.1895 | 710.4929 | 706.3115 | 795.7227 | 770.9069 |
| MNAD | 259.3633 | 258.0175 | 791.6371 | 788.2654 | 292.3340 | 293.4836 | 301.3480 | 301.2153 | 303.1933 | 310.9818 |
| **Ours** | 297.5692 | 330.1162 | 249.8632 | 263.4532 | 340.7552 | 363.7520 | 237.0070 | 234.7110 | 450.0926 | 721.3093 |

## D.2 Anomaly score

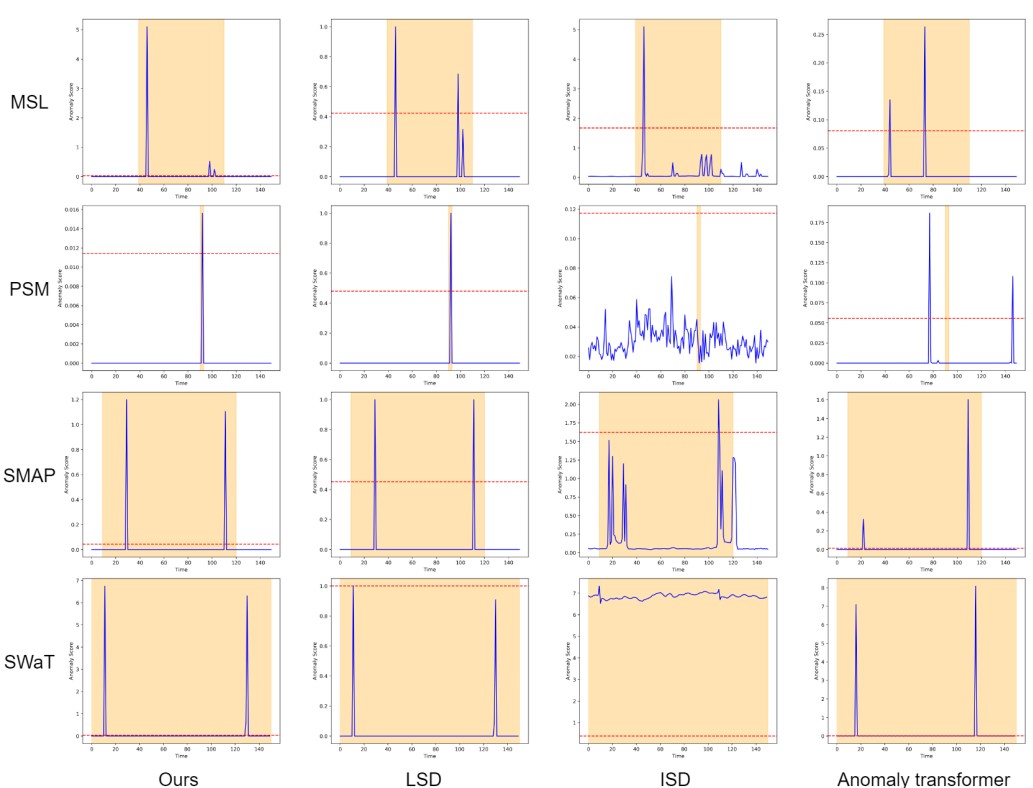

Figure 6: Visualization of anomaly scores for MSL, PSM, SMAP, and SWaT datasets.

Figure 6 visually represents the anomaly scores for benchmark datasets not discussed in Section 4.4. We randomly sampled data of length 150 from MSL, PSM, SMAP, and SWaT test datasets and plotted the anomaly scores for each segment. Compared to other baselines, our proposed method consistently detects anomalies precisely with a low false positive rate from the perspective of the point adjustment method.

