# OpenReview forum: "MEMTO: Memory-guided Transformer for Multivariate Time Series Anomaly Detection"
_NeurIPS.cc/2023/Conference — NeurIPS 2023 poster_

### Official Review · Reviewer_UfyT · 2023-06-10

**Soundness:** 3 good
**Presentation:** 3 good
**Contribution:** 3 good
**Rating:** 6
**Confidence:** 3

**Summary:**

This paper presents a reconstruction-based method for multivariate time series anomaly detection. It is a memory-guided Transformer which contains the gated memory module. Because of the training instability in updating memory items incrementally when the items are
initialized randomly, the authors propose a two-phase training paradigm to ensure stable training. The method calculates anomaly scores considering the input and latent space. The method achieves state-of-the-art results on five benchmark datasets.

**Strengths:**

1. The authors make a good explorations on leveraging memory module to multivariate time series anomaly detection task, and the proposed method can be effective solutions for the given task.
2. This paper contains enough experiment results and ablation study on five datasets to support its claims.
3. The proposed methodology is well presented, and most of the paper is well written.

**Weaknesses:**

1. The paper writing can still be improved. For example, the grammar error on line 88 "there exist video representation learning" -> "there exists video representation learning". Typo on line 234 "32.56p%". The citation in related work can be extended with author's name + 'et al.' rather than just a number.
2. The font size in the figures (in introduction and experiment) is pretty small.
3. I saw that you split the training data into 80% for training and 20% for validation. Do other methods for comparison in the experiments share the same data split(ratio and files)？ How many times you repeated your experiments? Since I saw the results you reproduced for Anomaly Transformer are quite different from their own paper, is it fair to list all these methods to compare with your model?


**Questions:**

Please address the points mentioned in Weakness.

**Limitations:**

Yes, the authors have addressed the potential negative societal impact of their work.

---

> ### Author Rebuttal · Authors · 2023-08-05
>
> We are thankful for your meticulous review and feedback on our paper. Your perceptive observations have steered us towards enhancing and refining our work.
>
>
> $\textbf{Weaknesses}$
>
> 1. We appreciate your keen attention to detail and will proceed to correct the grammar error on line 88, the typo on line 234, and the citation format as you pointed out.
>
> 2. We agree with your observation regarding the small font size in our figures. We will ensure to increase it for better readability as per your suggestion.
>
> 3. In our paper, all the methods including MEMTO employed the same data division ratio of 80% for training and 20% for validation. The main experiment in Table 1 in our paper was conducted once, but we carried out additional trials to calculate statistical measures such as averages and standard deviations, yielding robust scores over multiple runs (see Table 1 of the PDF, in the section on global response). Moreover, we noticed a discrepancy between the performance of the Anomaly Transformer reported in its original paper and our findings. While the original paper stated that they used 80% of the data for training and 20% for validation, a review of the official source code revealed a different setting. They used the entirety of the training data (100%) for training, inclusive of the 20% initially set aside for validation. Since this deviates from the standard measurement procedure, we decided to adjust the data settings for the Anomaly Transformer to align with that of MEMTO, by excluding the validation data from training. This adjustment ensures a fair comparison of performances.

---

> > ### Comment · Reviewer_UfyT · 2023-08-18
> >
> > My concerns have been addressed by authors.

---

> > > ### Author Response · Authors · 2023-08-18
> > > **Thank you for your comment**
> > >
> > > We are delighted that we could address all of your questions, and we deeply appreciate your active involvement.

---

### Official Review · Reviewer_EFum · 2023-06-26

**Soundness:** 2 fair
**Presentation:** 3 good
**Contribution:** 2 fair
**Rating:** 6
**Confidence:** 3

**Summary:**

The authors propose a memory-guided transformer with a reconstruction paradigm for multivariate time series anomaly detection. The time-series encoder, memory parameters of normal patterns, and projection heads are updated during training time in a two-step fashion. On test time, input queries are projected on the normal patterns learned base with a learned sparse linear projection. As a result, reconstructed anomalies tend to look like normal samples, amplifying the reconstruction loss and enabling detection. Experiments are run over common multivariate time-series datasets following previous works practices.

**Strengths:**

* The proposed gated memory module seems simple, original, and efficient for detecting anomalies on multivariate time series based on transformer encodings and input sample reconstruction.

* The authors provide thorough ablation studies and some insights into the importance of each part of their algorithm.

* The manuscript is overall easy to follow and well-written. Code is provided.

## Suggestions

The typical memory size is quite small, which I see as a positive point of this method. So, I suggest this could be emphasized in the main manuscript. I read the entire paper wondering what the memory size was and whether the trade-off memory-performance was good, just to be positively surprised in the appendix.

**Weaknesses:**

## Regarding novelty

1. Overall this work seems to be an incremental modification of [8], [24], and [40]. In particular, contribution #3 (line 77) does not seem very novel. I would like the authors to please discuss Equation (11) in this paper versus (6) in [40] in more detail.
2. How does the memory update stage is different from other memory gates, such as GRU, MRU, etc?

## Regarding state-of-the-art claim

1. Why not compare to MNDA [24], memAE [8], USAD[2] in Table 1? Why are A.T. numbers in Table 1 different from [40]? I would suggest adding error bars at least to these methods and yours to make the comparison fair. I know this was not done in these previous works, but claiming state-of-the-art without comparing confidence intervals in an empirical field is not a good practice.

## Regarding methodology

4. Not clear why not just have a memory on the input space and retrieve the closest samples instead of the proposed mechanism on the latent space?
5. Why is it so sensitive to k-means initialization? Is it computed with enough iterations of the K-means algorithm?
6. How does algorithm 1 outlines the two-phase training paradigm (line 175)? The first and second phase seems to be omitted.

## Other less important questions

7. How are the thresholds computed in figure 3?
8. What is the computation overhead when compared to A.T.? MEMTO seems simpler computationally. Could the authors discuss this?


**Questions:**

Please see the weaknesses section.

**Limitations:**

The authors have discussed potential limitations and shortcomings of their algorithm and no special negative societal impact exclusive of their work is evident.

---

> ### Author Rebuttal · Authors · 2023-08-06
>
> We're grateful for your detailed evaluation and insight on our research paper. We will make sure to incorporate the parts that you suggested for clarity and reflect their feedback on paper.
>
> $\textbf{Suggestion}$
> As you suggested, we will emphasize in the main manuscript that our proposed MEMTO is robust to the number of memory items and even performs well with small memory size (low resource setting), in our revised version.
>
> $\textbf{Regarding novelty}$
> 1. Drawing inspiration from [8], [24], and [40] like you mentioned, MEMTO stands out from prior works with the following distinctive features. Due to the limit on response length, we answered the follwoing question on the global response.
>
> 2. As you mentioned, both our proposed Gated memory module and other memory gates, such as the GRU and MRU, have the ability to control how much new information is injected. However, other memory gates, such as GRU and MRU, are often inadequately compartmentalized to retain precise knowledge from the past, as the information gets compressed into dense vectors. Therefore, it is impossible to have them compartmentalized and memorize diverse prototypical features of normal patterns in the data. In contrast, Gated memory update stage in our Gated memory module applies attention-based operation (query-conditioned memory attention) while writing new information into each memory item. Since each memory item can retrieve information from only relevant queries, this mechanism enables each memory item to be compartmentalized and retain prototypical features of normal patterns in the data.
>
> $\textbf{Regarding state-of-the-art claim}$
>
> 3. The reason we did not add MNAD [24] and memAE [8] as baselines in Table 1 in our paper is that they are anomaly detection models for the computer vision domain. The performance of MNAD and MemAE is shown in Table3 in our paper as the result of an ablation study on the memory module. In the experiment, only the memory module is modified while keeping all other experimental conditions, including the pipeline and architecture, identical to those used in MEMTO. Therefore, they cannot be considered as independent models for multivariate time series anomaly detection tasks, as the only difference lies in the memory module itself. Also, while USAD computes the anomaly score on a window level to determine if each window is anomalous, we compute the anomaly score on a timestamp level to ascertain the anomalousness of each time point. Therefore, we only used models that compute the anomaly score on a timestamp level as our baselines. Moreover, we agree that adding error bars is a valid way to properly demonstrate the state-of-the-art results and provide a more comprehensive understanding of the experimental outcomes. We provide you with some additional experiments comparing with the latest state-of-the-art model, Anomaly Transformer (A.T), on five real-world benchmark datasets.The results can be seen in Table 1 of the PDF, in the section on global response.
>
>
> $\textbf{Regarding methodology}$
>
> 4. The latent representations obtained from the transformer encoder for each timestamp consider the temporal dependency information between timestamps. On the other hand, the raw timestamp values in the input space do not inherently account for temporal dependencies with other timestamps, making them unsuitable for accurately learning the normal prototypical pattern of the time series. As you pointed out, not explicitly mentioning this in the main script could lead to ambiguity. We will clearly note this in our revised paper.
>
> 5. As mentioned in line 243~247, we empirically observed that updating memory items incrementally can result in training instability if the items are initialized randomly. Memory module initialization with K-means clustering enables injecting inductive bias of normal prototypical patterns into memory items. By providing the memory items with informative initial values (rough prototypical feature of the normal pattern), the model starts with a better understanding of the normal pattern at the second phase of two-phase training paradigm, which can lead to more stable and efficient learning compared to simply using random initialization. When experimented with multiple random seeds(using a two-phase training paradigm), MEMTO consistently demonstrates stable and robust detection performance. This result indicates that the iteration value, which is the hyper-parameter of the K-means algorithm, is large enough for our model to have reliable performance.
>
> 6. Thank you for careful checking of our paper. As you rightly pointed out, we will correct it as ‘Algorithm 1 outlines the memory module initialization with K-means clustering’ in our revised version.
>
> $\textbf{Other less important questions}$
>
> 7. In the case of threshold selection, you can refer to Appendix A Training details (line 440~442).
>
> 8. The computation analysis between MEMTO and Anomaly Transformer is explained in the response on the first question (Regarding novelty). To quantitatively verify the computational efficiency in MEMTO, we conduct additional experiments for inference time of MEMTO compared to Anomaly Transformer on five real-world benchmark datasets. Results in Table 4 of the PDF in the section on global response, show that whereas training time for MEMTO is longer than that of Anomaly Transformer, its inference time in real time detection process is shorter.
> Since we agree that computation overhead is crucial measurement for real-world application, we will include this content in the revised version.

---

> > ### Comment · Reviewer_EFum · 2023-08-11
> >
> > Thank you for the rebuttal and the additional experiments. All my concerns were addressed, especially regarding the experimental and novelty parts. I raise my recommendation accordingly.

---

> > > ### Author Response · Authors · 2023-08-16
> > > **Thank you for your comment**
> > >
> > > We are grateful to you for dedicating time to read our clarifications and for reevaluating your assessment of our work.

---

### Official Review · Reviewer_C8uM · 2023-07-01

**Soundness:** 3 good
**Presentation:** 3 good
**Contribution:** 3 good
**Rating:** 7
**Confidence:** 4

**Summary:**

The paper used memory network to capture frequently present normal pattern in date set. While training, the memory are built in latent space in the autoencoder framework.  Along with reconstruction error, the distance between a representation with the closest memory unit is used to calculate anomaly score. If a data does not have representation close to memory then that is an anomaly.


**Strengths:**

Idea of keeping more than one memory for common patterns from normal data is Novel and makes more sense than existing methods of keeping memory of anomalies.  Experimental results were produced on a large number  of data sets.

**Weaknesses:**

Literature surveys missed graph based anomaly detection techniques for multivariate time series like Deng, Ailin, and Bryan Hooi. "Graph neural network-based anomaly detection in multivariate time series." In Proceedings of the AAAI conference on artificial intelligence, vol. 35, no. 5, pp. 4027-4035. 2021.


It is required to have an ablation study in the main paper to bring our significance of memory gate and Kmeans clustering.


**Questions:**

Most cases  the F1 score for the proposed method is close to the A.T benchmark. Did you do a statistical significance test?
Do you run your experiment for different random train text split? In that case how is the  variance of accuracy ?
The F1 score for only LSD and only ISD is quite low. Did you note down if there are more FP or FN in both cases?
How to decide the required number of memory unit ?


**Limitations:**

Number of majority patterm in normal part need to be known before which is not possible during unsupervised task.

---

> ### Author Rebuttal · Authors · 2023-08-05
>
> We are grateful for your careful reading of our paper, and your feedback. Your insightful feedback has guided us in expanding and improving our paper.
>
> $\textbf{Weaknesses}$
>
> $\textbf{Literature surveys missed graph based anomaly detection techniques for multivariate time series}$
>
> We will also include graph-based anomaly detection methods for multivariate time series such as the paper (1), (2), and (3) in the literature review.
>
> (1)   Deng, Ailin, and Bryan Hooi. "Graph neural network-based anomaly detection in multivariate time series." In Proceedings of the AAAI conference on artificial intelligence, vol. 35, no. 5, pp. 4027-4035. 2021.
>
> (2)   Zhang, Weiqi, Chen Zhang, and Fugee Tsung. "Grelen: Multivariate time series anomaly detection from the perspective of graph relational learning." Proceedings of the Thirty-First International Joint Conference on Artificial Intelligence, IJCAI-22. Vol. 7. 2022.
>
> (3)   Zhao, Hang, et al. "Multivariate time-series anomaly detection via graph attention network." 2020 IEEE International Conference on Data Mining (ICDM). IEEE, 2020.
>
> $\textbf{It is required to have an ablation study in the main paper to bring our significance of memory gate and Kmeans clustering.}$
> The key aspect of the Gated memory module is the update gate, which is the most significant point of differentiation compared to MNAD and MemAE. Therefore, to assess the capability of our proposed Gated memory module mechanism, we conduct an ablation study comparing MEMTO with other memory modules in Section 4.3 (line 232-242) and present the following results in Table 3. Moreover, as for ablation study on memory module initialization with K-means clustering, experimental results in Table 3 provide strong evidence of its effectiveness (see section 4.3 Ablation study line 243-249).
>
>
> $\textbf{Questions}$
>
> $\textbf{Most cases the F1 score for the proposed method is close to the A.T benchmark. Did you do a statistical significance test?}$
> We had not performed statistical significance tests in the manuscript since it was not generally done in the previous works. However, as you requested, we conduct additional experiments with varying random seeds to verify the mean and standard deviation of the performance  (see Table 1 of the PDF, in the section on global response). We also perform statistical tests comparing our results to the latest state-of-the-art model, Anomaly Transformer. We conduct a t-test to prove a significant performance difference between MEMTO and Anomaly Transformer. The results show a p-value less than 0.05 across all datasets, confirming a significant performance difference between the two models (see Table 1 of the PDF, in the section on global response).
>
> $\textbf{Do you run your experiment for different random train text split? In that case how is the variance of accuracy?}$
> We did not run the experiment for different random train-test splits. The reason is that the train dataset and test dataset from the benchmarks are pre-split and designated when downloaded from the archive, where the anomaly labels exist only in the test dataset.
>
> $\textbf{The F1 score for only LSD and only ISD is quite low. Did you note down if there are more FP or FN in both cases?}$
> In Table 2 in our paper, we provided F1 score as the only performance metric. Nonetheless, at our experiments, we also record precision and recall values. The results we obtained can be seen in Table 2 of the PDF, in the section on global response. Both precision and recall values are lower in the "Only LSD" or "Only ISD" cases compared to the "LSD+ISD" case. This indicates that in both cases, we have more false positives (FP) and false negatives (FN) compared to the combined approach (LSD+ISD).
>
> $\textbf{How to decide the required number of memory unit?}$
> In the Supplementary material, Appendix C.2 (line 478), there is an experiment result conducted on the number of memory items. Through this experiment, we discovered that MEMTO is robust to the number of memory items and works effectively even with a small number of items. Considering computational efficiency, we chose to use only 10 memory items as default value in our experiments.
>
>
>
>
> $\textbf{Limitations}$
>
> $\textbf{Number of majority patterm in normal part need to be known before which is not possible during unsupervised task.}$
> The number of memory items, identical to the number of prototypical patterns of normal data, is not a known prior value but a hyperparameter. We experimented by varying this hyperparameter in Appendix C.2 and found that MEMTO shows robust performance regardless of its value.

---

> > ### Comment · Reviewer_C8uM · 2023-08-13
> >
> > I thank Author for answering all my questions.

---

> > > ### Author Response · Authors · 2023-08-16
> > > **Thank you for your comment**
> > >
> > > We are pleased that all of your questions have been resolved, and we are grateful for your engagement.

---

### Official Review · Reviewer_pZZS · 2023-07-05

**Soundness:** 2 fair
**Presentation:** 2 fair
**Contribution:** 1 poor
**Rating:** 4
**Confidence:** 4

**Summary:**

The authors of this paper focused on tackling the problem of over-generalization in reconstruction-based deep models and made a contribution to the field of multivariate time series anomaly detection. The main challenge they encountered was dealing with complex dependencies and inter-variable correlations within the data. To address this, they proposed a novel gated memory module approach. This module learns how much each memory item should be updated based on the input data, effectively capturing the prototypical features of normal patterns in the data. By doing so, it aims to overcome the inability of existing models to capture dynamic nonlinear temporal dependencies and complex inter-variable correlations.

**Strengths:**

• The authors explained the importance of multivariate time series anomaly detection with the proper real-world examples
• Comprehensive comparison of their approach with traditional methods (OC-SVM, Deep-SVDD, LSTM-VAE, Anomaly Transformer etc.)
• Thorough explanation of the training hyper-parameters, dataset setting, algorithms and code
• Achieved state-of-the-art results on five real-world benchmark datasets

**Weaknesses:**

• Lack of theoretical proof: two-phase training approach and bi-dimensional deviation-based criterion need a detailed theoretical proof
• Limited discussion of computational efficiency: how gated memory module and two-phase train- ing affect the training time
• I couldn’t find any explicit explanations that how the LSD and ISD help the training. In the descriptions, LSD and ISD only appears in Table2 (and the sec3.4). If LSD+ ISD just help the anomaly score but doesn't help the training, this should be clearly justified.
• Need more quantitative experiment about the improvement from gated memory seems limit. (Table 3)

**Questions:**

• How does the proposed method compare to existing approaches in terms of computational efficiency? Were any considerations or optimizations implemented to ensure its feasibility in real- time or resource-constrained environments?
• What led the authors to choose K-means clustering for initializing memory items? Is K-means considered the most effective approach for this task, particularly in terms of computational efficiency?
• How does the model behavior change if sub-series are generated with the help of overlapping sliding window?
• Since K-means clustering creates prototypes based on the seen data items, how does the model handle unseen cases? If the proposed MEMTO couldn't consider the degree of new information being injected into existing memory items, what is the contribution of the MEMTO compared to the MNAD?

**Limitations:**

Please refer to the Weaknesses and Questions.

---

> ### Author Rebuttal · Authors · 2023-08-05
>
> Thanks for your dedicated review of our work. Your critical feedback have helped us to extend and refine the paper. We provide a detailed response to your comments.
>
> $\textbf{Weakness}$
>
> - $\textbf{Lack of theoretical proof}$: We agree that we lack theoretical proof, and we have mentioned this in Conclusion(line 286). We plan to address this as a topic of our future research.
> - $\textbf{Limited discussion of computational efficiency}$: As you suggested, we measured the training time with and without using Gated memory module and the two-phase training paradigm, respectively. The results are presented in Table 4 of the PDF, in the section on global response. Training MEMTO without the Gated memory module reduces both the number of parameters and training time, but it leads to decreased performance by 26.27%p. Also, while the two-phase training paradigm requires approximately 2.45 times more time for training compared to its absence, the inference time remains the same. As shown in Table 3 in our paper, using a two-phase training paradigm improves the performance by 10.4%p, leading us to apply it at the cost of increase in training time.
> - $\textbf{Explanation of LSD and ISD}$: As you can see in Section 3.4, LSD and ISD are used in our proposed anomaly score computation scheme, which reflects how anomalous a given timestamp is. As conventionally done in anomaly detection task, anomaly score is only for inference and is completely irrelevant to the training process. Table 2 in our paper represents an ablation study where we experimented with varying methods of calculating the anomaly score while performing inference on the test dataset with MEMTO.
> - $\textbf{Quantitative experiment of Gated memory module}$: The 'Statistical analysis of LSD' experiment conducted in Section 4.4 is related to the Gated memory module (See line 251). To verify if the Gated memory module effectively captures the prototypical normal patterns of the data, we evaluated the distances from the memory items within the Gated memory module to queries of normal timestamps and to queries of abnormal timestamps, respectively.
>
> $\textbf{Questions}$
>
> - $\textbf{Computational efficiency of MEMTO / Considerations for real-world applications}$: We compare the computational efficiency between MEMTO and the previous state-of-the-art model, Anomaly Transformer. The Anomaly Transformer computes and retains series-association and prior-association per encoder layer, then averages the association discrepancies across layers. Performing this process during inference is computationally demanding, while MEMTO, requiring only dot product operations between each query and a few memory items, is simpler and results in shorter inference time. We conducted an additional experiment to verify the effectiveness of MEMTO compared to the previous state-of-the-art model in the aspect of inference time. The results can be seen in Table 4 of the PDF, in the section on global response. The resources used to train MEMTO only require four GTX 1080 GPUs, and the training time took about 2 minutes on SWaT dataset, which can be considered as low computation resources and short training time compared to recent deep learning models. To ensure fast and effective operation in real world applications, we utilized only ten memory items within the Gated memory module and deployed just two MLP layers in the decoder, which makes the little number of parameters possible.
> - $\textbf{Reasons to use K-means clustering for initializing memory items}$: Since our objective for initializing memory items was to quickly acquire rough normal prototypical patterns rather than precisely, we chose K-means clustering, which has relatively low computational cost. Given $n$ as the number of data points, the time complexity of K-means clustering is $O(n)$, whereas that of Hierarchical clustering or Expectation Maximization clustering is both $O(n^3)$. Considering computational efficiency, we ultimately opted for K-means clustering.
> - $\textbf{Overlapping sliding window vs non-overlapping sliding window}$: We did not employ an overlapping sliding window when constructing sub-series due to computational inefficiency during the model's training and inference phase. Using an overlapping sliding window increases the size of the training dataset, which in turn augments the computational cost required to train the model. Also, when performing inference on the test dataset, the computational cost increases as the anomaly score for overlapping timestamps needs to be calculated multiple times and averaged.
> We observe the following changes in performance when using an overlapping sliding window (see Table 3 of the PDF, in the section on global response).
> - $\textbf{How does the model handle unseen cases during memory module initialization with K-means clustering?}$: For K-means clustering, we randomly sampled 10% of the training data under the assumption of iid (independent and identically distributed). Given this assumption, we believe the distribution of the remaining 90% of the training dataset, not used for clustering, is similar to that of the sampled 10%.
> - $\textbf{The contribution of MEMTO compared to MNAD}$: We agree that the structure of the Gated memory module in MEMTO is similar to the memory module in MNAD, if we set aside the ability to control the degree of injecting new information into existing memory items. However, MEMTO simplifies the anomaly score computation process with bi-dimensional deviation-based detection criterion, which consists of multiplication of normalized LSD and ISD, where LSD functions as weights for amplifying the normal-abnormal gap in ISD. More detailed explanations regarding this question are elaborated on global response.

---

> > ### Comment · Reviewer_pZZS · 2023-08-18
> >
> > I would like to thank the authors for the additional information in the rebuttal. I have decided to adjust my score to "Borderline reject" as most of my concerns have been addressed. Nonetheless, I remain uncertain about the choice of using K-means clustering for initialization due to its computational efficiency. I am not yet convinced that this is the optimal approach, which is why I refrained from elevating my score to the acceptance level.

---

> > > ### Author Response · Authors · 2023-08-18
> > > **Response to Reviewer pZZS**
> > >
> > > Dear Reviewer pZZS,
> > >
> > > We would like to express our gratitude for taking the time to carefully read our rebuttal.
> > >
> > > As mentioned in the rebuttal, we propose a two-stage training paradigm that uses a clustering method to set the initial value of memory items to the approximate normal prototypical pattern of the data. As you pointed out, there is no guarantee that 'K-means clustering' is the optimal method. Our claim is not that 'K-means clustering is the optimal choice among various clustering methods,' but rather the effectiveness of a 'two-phase training paradigm that allows for the selection of an appropriate clustering algorithm depending on the type of dataset.' Table 3 in our manuscript shows that designating the initial value of memory items through this 'two-phase training paradigm' is highly effective. According to the experimental results for five real-world datasets, the performance of MEMTO trained with the two-phase training paradigm applied outperforms the performance without the two-phase training paradigm by an average of 8.4%p. Specifically, in the SMAP dataset, the difference is 26.54%p.

---

### Author Rebuttal · Authors · 2023-08-06

We are very grateful to all the reviewers for your careful reading of our paper and helpful feedback. We will make sure to incorporate the parts that you suggested for clarity and reflect your feedback on the revised paper. We have compiled the results of additional experiments related to the reviewers' questions and comments in the attached PDF, and kindly ask you to review them. The following responses address common questions raised by several reviewers.

$\textbf{Comparing MEMTO with the Anomaly Transformer [40] in the aspect of anomaly criterion}$

Eq (11) in our paper and eq (6) in [40] are the equations for calculating anomaly scores for real-time detection process (inference). Since both belong to reconstruction-based models, they use reconstruction error as the main criterion for distinguishing between normal and abnormal samples. However, the LSD in MEMTO and the Association Discrepancy in Anomaly Transformer represent fundamentally different concepts. LSD, the distance between each time point’s query and its nearest memory item in latent space, employs the memory items’ nature of storing prototypical features of normal patterns in the data. However, Association Discrepancy, quantified by the distance between each time point’s prior-association and its series-association, utilizes the inherent normal-abnormal distinguishability of the association distribution. The Anomaly Transformer calculates and stores series-association and prior-association in memory for each encoder layer, and then averages the association discrepancies across multiple layers. Performing this process during inference is computationally demanding, while MEMTO, requiring only dot product operations between each query and a few memory items, is simpler, resulting in shorter inference time.

$\textbf{Comparing MEMTO with MemAE [8] and MNAD [24] in the aspect of overall architecture}$

Previous works [8] and [24] are both for the anomaly detection task in the computer vision domain. We proposed MEMTO, an algorithm that takes the characteristics of time series data into account and appropriately applies a memory module to the anomaly detection task in the time series domain for the first time. You can refer to Section 2 for comparison with [8] and [24] in the aspect of memory module mechanism. In particular, our proposed Gated memory module’s main distinction from [8] and [24] is written in line 93~97. Also, to further alleviate the over-generalization problem in reconstruction-based models, we use a weak decoder which requires a small number of parameters. While existing models, including both [8] and [24], typically use a symmetrical encoder-decoder architecture, we deviate from this norm by adopting an asymmetric model structure with a weak decoder. Leveraging a decoder with less parameters also enables our MEMTO to be much more computationally efficient.

---

### Decision · Program_Chairs · 2023-09-21

**Decision:**

Accept (poster)

**Comment:**

This paper addresses a method for overcoming the over-generalization which typically occurs in the reconstruction-based methods for anomaly detection. The approach taken in this paper is to employ the memory module where each memory is updated in response to the input data in such a way that the over-generalization is alleviated. Two main contributions were claimed, including the gated memory module and the two-phase training with k-means clustering for initialization. In fact, there are a few memory-augmented methods, but this work might be the first attempt for applying such an idea to time series, which is valuable. Most of reviewers agree that the method in this paper is sound. Most of concerns raised by reviewers were well responded. However, one reviewer had some concerns remained even after the rebuttal. He/she was not convinced with the k-means clustering for initialization. I think using k-means to initialize prototype vectors is a minor issue, which anyone can think of as a baseline technique.  If authors claims that it is the one of main contributions in this paper, I doubt it. Despite some arguments, the paper has some interesting contributions that can be presented at the conference.